# Responses of Three *Pedicularis* Species to Geological and Climatic Changes in the Qinling Mountains and Adjacent Areas in East Asia

**DOI:** 10.3390/plants13060765

**Published:** 2024-03-08

**Authors:** Qijing Zhang, Zhaoping Lu, Mingchen Guo, Jia Kang, Jia Li, Xiaojing He, Jiayi Wu, Ruihang Liu, Jiaxin Dang, Zhonghu Li

**Affiliations:** Key Laboratory of Resource Biology and Biotechnology in Western China, Ministry of Education, College of Life Sciences, Northwest University, Xi’an 710069, China; zhangqijing@stumail.nwu.edu.cn (Q.Z.); 202032601@stumail.nwu.edu.cn (Z.L.); 13633436166@163.com (M.G.); kangjia202309@163.com (J.K.); lejea2020@163.com (J.L.); heexj13@163.com (X.H.); wujiayi5p1@163.com (J.W.); 15869926945@163.com (R.L.); 19909292608@163.com (J.D.)

**Keywords:** divergence, geographic distribution, evolutionary relationship, ecological niche analysis, *Pedicularis*

## Abstract

The Qinling Mountains in East Asia serve as the geographical boundary between the north and south of China and are also indicative of climatic differences, resulting in rich ecological and species diversity. However, few studies have focused on the responses of plants to geological and climatic changes in the Qinling Mountains and adjacent regions. Therefore, we investigated the evolutionary origins and phylogenetic relationships of three *Pedicularis* species in there to provide molecular evidence for the origin and evolution of plant species. Ecological niche modeling was used to predict the geographic distributions of three *Pedicularis* species during the last interglacial period, the last glacial maximum period, and current and future periods, respectively. Furthermore, the distribution patterns of climate fluctuations and the niche dynamics framework were used to assess the equivalence or difference of niches among three *Pedicularis* species. The results revealed that the divergence of three *Pedicularis* species took place in the Miocene and Holocene periods, which was significantly associated with the large-scale uplifts of the Qinling Mountains and adjacent regions. In addition, the geographic distributions of three *Pedicularis* species have undergone a northward migration from the past to the future. The most important environmental variables affecting the geographic distributions of species were the mean diurnal range and annual mean temperature range. The niche divergence analysis suggested that the three *Pedicularis* species have similar ecological niches. Among them, *P. giraldiana* showed the highest niche breadth, covering nearly all of the climatic niche spaces of *P. dissecta* and *P. bicolor*. In summary, this study provides novel insights into the divergence and origins of three *Pedicularis* species and their responses to climate and geological changes in the Qinling Mountains and adjacent regions. The findings have also provided new perspectives for the conservation and management of *Pedicularis* species.

## 1. Introduction

Climatic oscillations during the Quaternary have profoundly affected the geographic distribution, migration, and genetic structure of plants in both the Southern and Northern Hemispheres [1,2]. Although East Asia was not covered much by ice sheets during the last glacial period, climate oscillations during the late Quaternary strongly impacted the biological diversity and evolution of many extant species in the region [3,4]. Recent phylogeographic studies have shown that climatic changes and historical geological events were the primary drivers of population expansions/contractions and interspecific divergence in plants, which also constructed new ecological niches, leading to the origin and divergence of new species [5,6]. In addition, the combination of ecological niche modeling (ENM) and maternal inherited chloroplast DNA (cpDNA) markers were used to predict the geographic distribution changes of relict plant species and forest trees from East Asia [7,8]. Combining plant genetic information with niche models could help us investigate the effects of global climate change on the evolution of alpine plants since the Quaternary, as well as the factors involved in the origin and differentiation of species [9,10,11]. These studies provided novel insights into biodiversity conservation in the context of global climate.

Previous geological studies suggested that the Qinling Orogenic Belt in East Asia experienced a long evolutionary history and has a complicated composition and structure [12]. The rapid uplift of the Qinling Mountains since the late Cenozoic was influenced by changes in deep earth dynamic processes, which dominated the different climatic environments in mountainous areas and subsequently created the abundant biodiversity of the Qinling Mountains [13,14]. The Qinling Mountains are a unique biodiversity region in the world due to their distinctive geographic location, diverse habitats, and climate types, providing an ideal environment for studying species differentiation and response patterns of plant species to geological and climatic oscillations. Relevant studies have investigated the genetic divergence and evolutionary histories of three closely related tree peony species (*Paeonia qiui* Y. L. Pei and D. Y. Hong, *Paeonia jishanensis* T. Hong and W. Z. Zhao, and *Paeonia rockii* (S. G. Haw and Lauener) T. Hong and J. J. Li ex D. Y. Hong) in the Qinling–Daba Mountains [5] and also revealed the population genetic structure of *P. rockii* using nuclear gene markers and chloroplast DNAs [15]. These results demonstrated that the Qinling Mountains as a geographic barrier have profoundly impacted the population evolution of the endemic species in this region. However, the phylogeographic studies and population demographic history of endemic plants in the Qinling Mountains and adjacent areas need further exploration.

*Pedicularis* (Orobanchaceae) is a genus of hemiparasitic plants distributed throughout the Northern Hemisphere, mainly in high-latitude or montane habitats. The species richness and morphological diversity in *Pedicularis* is confined to a comparatively limited region of eastern Asia, with approximately 75% of species being endemic to the Himalaya-Hengduan areas [16,17]. The genus *Pedicularis* provides an exceptional model or case in this region for exploring how alpine plants respond to climatic fluctuations and geological events, as well as the associated interspecific evolutionary processes and population dynamics [18,19]. Recently, the evolutionary patterns of three single/low-copy nuclear genes (*CRC*, *LFY-L*, *LFY-S*) and two chloroplast genes (*matK* and *ycf1*) in sixty-five accessions of *Pedicularis* sect. *Cyathophora* from the Hengduan Mountains were investigated, which revealed that the uplift of the Qinghai–Tibet Plateau and climatic changes may have played crucial roles in the divergence and speciation of *Pedicularis* sect. *Cyathophora* [20]. Moreover, the effects of climatic oscillations on genetic diversity and phylogenetic relationships in *Pedicularis* sect. *Cyathophora* (Orobanchaceae) on the Qinghai–Tibet Plateau have been also researched [21,22]. Nevertheless, the evolutionary relationships and species divergence of *Pedicularis* species in the Qinling Mountains and adjacent areas are less well known. Therefore, we focus on three species of *Pedicularis* (*Pedicularis bicolor* Diels, *Pedicularis dissecta* (Bonati) Pennell and H. L. Li, *Pedicularis giraldiana* Diels ex Bonati) endemic to the Qinling Mountains [23]. *P. bicolor* has been recorded as a vulnerable (VU) species on the IUCN Red List [24]. *P. giraldiana* is distributed over most parts of the Qinling Mountains and extends southwestward in China ranging from 2900 to 3000 m, while *P. dissecta* is distributed in the Qinling Mountains of southern Shaanxi province, thriving on rocks at an altitude of 3000 m [23].

In this study, we analyzed phylogenetic relationships and divergence of three *Pedicularis* species (*P. bicolor*, *P. giraldiana,* and *P. dissecta*) using maternally inherited chloroplast DNA (cpDNA) sequences. Additionally, we examined niche divergences between the species through ecological niche modeling (ENM) and compared their spatial distributions during different historical times. The main objectives of this study were to (1) conduct phylogenetic analysis based on plastid genome data; (2) estimate divergence times and explore their relationships with geological events; (3) evaluate niche differences between species from both geographic and environmental perspectives; (4) investigate the impacts of global climate change on the three *Pedicularis* species in the Qinling Mountains since the last interglacial (LIG) period and explore their response to climate oscillations and geological events; and (5) provide references for protecting wild plant population resources.

## 2. Results

### 2.1. Phylogenetic Relationship

The phylogenetic relationship of *Pedicularis species* was reconstructed based on maximum likelihood (ML) and maximum parsimony (MP) analyses using the chloroplast genomes of seventeen Orobanchaceae and three Lamiaceae species. Overall, the topologies obtained using different methods (ML and MP analyses) were almost identical. It was shown that all examined Orobanchaceae species formed a monophyletic evolutionary clade with high bootstrap support (Figure 1). Within this major evolutionary clade, *Pedicularis* and *Orobanche* were found to cluster sister relationships. Each of the two clades formed a monophyletic branch with a 100% bootstrap value, respectively. *P. dissecta* and *P. bicolor* are both nested branches within *Pedicularis*; they have high support and are nested within the larger *Pedicularis* evolutionary branch. *Pedicularis cheilanthifolia* Schrenk and *P. giraldiana* are grouped together in a single branch with strong support from bootstrap values.

### 2.2. Divergence Time Estimation

Based on the chloroplast genomes, the divergence time estimation indicated that the two large clades, *Pedicularis* and *Orobanche*, diverged approximately 47.7 million years ago (Figure 2). According to the results, the crown age of the diversification of *Pedicularis* was estimated to be around 35.9 Ma (95% highest posterior density, HPD: 20.9–45.0 Ma). The two sister species in the genus *Pedicularis* comprising *P. giraldiana* and *P. cheilanthifolia* diverged by about 0.4 Ma (95% HPD: 0.1–1.7 Ma). The crown ages of *P. bicolor* and *P. dissecta* took place at approximately 13.1 Ma (95% HPD: 5.2–23.5 Ma) and 12.9 Ma (95% HPD: 4.1–19.3 Ma), respectively.

### 2.3. Environmental Variables Analysis

By correlating the factors and geographical data and combining the contribution amounts, eight climatic variables, including annual mean temperature (bio1), mean diurnal range (bio2), temperature annual range (bio7), mean temperature of the wettest quarter (bio8), mean temperature of the driest quarter (bio9), mean temperature of the warmest quarter (bio10), mean temperature of the coldest quarter (bio11), and precipitation of the warmest quarter (bio18), were finally selected for modeling in this study (Figure 3). The current potential distribution of *P. bicolor* is attributed to the bio1, bio2, and bio7 variables, which account for 20.3%, 35.9%, and 13.8%, respectively, resulting in a cumulative contribution percentage of over 60%. The three most significant contributors to the currently suitable habitats of *P. dissecta* are bio2, bio9, and bio10, accounting for 32%, 14.7%, and 24.3%, respectively, and representing 71% of the total. The contribution variables of *P. giraldiana* are bio2, bio7, and bio18, accounting for 37.9%, 13.9%, and 27.8%, respectively. These three environmental factors accounted for more than 70% of the total. In summary, the range of diurnal and annual temperature changes plays a crucial role in influencing the geographic distributions of three *Pedicularis* species.

### 2.4. Dynamic Changes in Geographic Distribution from the Past to the Future

Based on the effective specimen site information, the ecological niche model was used to simulate the past, current, and future niche distributions of three *Pedicularis* species. The areas under curve (AUC) values of each species model test above 0.90, indicating that the prediction results of the potential distribution area of the species were credible. During the LIG, the three *Pedicularis* species occurred, mainly in southwest China (eastern Sichuan, southern Chongqing, Guizhou, and northern Guangxi) (Figure 4). In contrast to the LIG period, the potential distributional ranges of three *Pedicularis* species shrank to Shaanxi, Sichuan, and southeastern Gansu province during the last glacial maximum (LGM) period. From the LGM to the current period, the suitable habitat of three *Pedicularis* species has rapidly increased. It appeared that each species gradually migrated northward from the LIG to the current period.

The results of the future distributions (2070s) showed a significant reduction in suitable habitats for three *Pedicularis* species compared to the present times, especially highly suitable habitats, which will shrink to southern Shaanxi under the representative concentration pathway 2.6 (RCP 2.6) scenario. Additionally, it is worth noting that the distributional ranges of suitable habitats for *P. bicolor* and *P. dissecta* will decrease significantly from 2050 to 2070 under the RCP8.5 scenario (Appendix A).

### 2.5. Migration Trends Based on the Centroid of Suitable Habitats

Under various climatic scenarios, the centroid of each species is predicted to migrate from the south (northern Guizhou) to the north of China (northern Sichuan and southern Shaanxi) (Figure 5). For instance, it was shown that the three *Pedicularis* species had undergone a northward migration from the LIG to LGM periods; while the two species of *P. bicolor* and *P. dissecta* have shifted to the northeast of China, the geographic ranges of *P. giraldiana* moved to the northwest of China. Under the RCP2.6 scenario, *P. bicolor* and *P. dissecta* initially migrated southeast and northwest of China for a short distance, respectively, and then both of them migrated northeast of China in 2070. In the RCP8.5 scenario, all three species will eventually migrate to the east or northeast of China over time.

### 2.6. Niche Analysis

In the niche comparison analysis, the first two axes explained 88.9% of the total variation of climatic conditions for the *Pedicularis* species ranges (PC1 = 70.3%, PC2 = 18.6%) (Figure 6). The multiple niche plot illustrating the 20% of occurrence density showed that the realized niche of *P. bicolor* and *P. dissecta* were the most closely related, whereas *P. giraldiana* was the most widely realized niche and contained both *P. bicolor* and *P. dissecta*. When 100% of the occurrence density was plotted in environmental space, a high degree of climatic spatial overlap was detected between the ranges. All pairwise ecological niche modeling comparisons produced high values for Hellinger’s *I* and Schoener’s *D* and displayed high degrees of geographic overlap (Figure 7).

Moreover, *P. giraldiana* showed a highest niche breadth, covering almost all climatic niche spaces of *P. dissecta* and *P. bicolor*. In the two multiple niche PCA-env plots, it is evidenced that *P. bicolor* had the smallest niche breadth compared to the other species. Additionally, *P. bicolor* and *P. dissecta* occupied the drier and colder climatic niches, while *P. giraldiana* occupied a warmer and wetter niche.

A pairwise comparison of the climatic niches of three species of *Pedicularis* showed that *P. bicolor* and *P. giraldiana* have the lowest environmental niche overlap, while *P. dissecta* and *P. bicolor* have the highest niche overlap. The climatic niche classification results indicated that the “Unfilling” ecological niche accounted for 0–0.126, the “Stability” niche accounted for 0.276–0.624, and the “Expansion” niche accounted for 0.376–0.724 (Table 1).

## 3. Discussion

### 3.1. Evolutionary Relationships

In recent years, an increasing number of studies have used chloroplast genome data for species phylogeny and interspecific differentiation studies to explore the origin and evolution of species [25]. The frequent parallel or convergent evolution of characters of corollas has resulted in the difficulties of the genus *Pedicularis* classification [26]. Traditionally, *Pedicularis* was placed in the Scrophulariaceae, but molecular evidence showed that this genus has been shifted to Orobanchaceae and found to be polyphyletic [27,28,29]. Furthermore, there was little consensus between the phylogenetic tree and conventional classification in *Pedicularis* [30]. Yu et al. (2015) constructed a backbone phylogeny of *Pedicularis,* with extensive species sampling from the Himalaya–Hengduan Mountains, using sequences of the nuclear ribosomal internal transcribed spacer (nrITS) and three plastid regions (*matK*, *rbcL*, and *trnL-F*) [31]. Liu et al. (2022) reconstructed evolutionary relationships of the *Pedicularis siphonantha* Don complex using nrITS and four plastid DNA loci (*matK*, *rbcL*, *trnH-psbA*, and *trnL-F*) to resolve taxonomic confusion [32].

Our results showed that two major evolutionary clades were identified comprising *Pedicularis* and *Orobanche* with high bootstrap values (Figure 1). In the first major clade, *P. giraldiana* was placed as a sister to *P. cheilanthifolia* with high bootstrap value support. The lack of consensus between the phylogenetic tree and previous studies may be explained by the small sample size of *Pedicularis* species in previous studies. In agreement with previous studies, *Pedicularis verticillata* L. and *Pedicularis hallaisanensis* Hurus. have clustered the sister evolutionary clades [33]. Another phylogenetic study showed that *Pedicularis* species clustered together and *Pedicularis shansiensis* P. C. Tsoong and *P. dissecta* formed a clade based on complete chloroplast genomes [34]. Additionally, some studies showed that *P. dissecta* and *P. bicolor* were more closely related to *P. shansiensis*. However, our study suggested that these two species formed nested relationships, possibly due to the differences in DNA molecular markers used in different studies. The intrageneric phylogenetic relationships of many studies were less consistent based on the different molecular markers, presumably related to pseudogenization and gene loss (*accD* and *ccsA*) in *Pedicularis* species [35]. In addition, Yang et al. (2007) detected an unusually high sequence divergence among congeners in a comparatively small geographical range, as well as frequent large deletions in the cpDNA *trnT-trnF* region of *Pedicularis* species [36]. To verify species relationships and interspecific divergence in *Pedicularis*, it will be crucial to collect more species samples in future research.

### 3.2. Species Divergence

It is well known that dated phylogeny is essential for the comprehension of speciation processes [37]. Our molecular dating results suggested that the crown age of the diversification of *Pedicularis* species was estimated to be around 35.9 Ma (Figure 2). This result was consistent with a recent study and was supported by the earliest pollen fossil record of *Pedicularis* from the Eocene in China [20,35,36,37,38,39,40,41]. According to our molecular dating analysis, *P. dissecta* and *P. bicolor* may have occurred during the late Miocene (approximately 13.1 Ma and 12.9 Ma, respectively). This timing is closely associated with the onset of a new period of tectonic thermal activity and uneven vertical uplift in the Qinling Mountains since the Miocene [12,42], which might have played crucial roles in the divergence of *P. dissecta* and *P. bicolor*.

Interestingly, the divergence of *P. giraldiana* and *P. cheilanthifolia* was dated approximately 0.4 Ma by secondary calibration. Yuan et al. (2011) have also investigated the population divergence of *Paeonia rockii*, a species endemic to the Qinling Mountains and the adjacent area, using cpDNA sequences [15]. The results showed that species divergence (0.4 to 1.6 Ma) was concurrent with the time scale of the latest rapid uplifts of the Qinling Mountains during the Pleistocene (approximately 0.7 Ma) [43,44]. Previous studies have shown that the diversification of *Pedicularis cyathophylloides* Limpr. during the middle Quaternary (0.4–2.5 Ma) was estimated by *LFY-L* nuclear genes [20]. Additionally, it was discovered that the section *Cyathophora* of *Pedicularis*, which was only found in the Sino–Himalayan regions, formed a well-supported monophyletic group. The species divergence of this section was largely associated with mountain uplifts and geological oscillations [36]. In summary, in order to further verify the affinities between species, more reliable fossils are required for phylogenetic studies, and data from multiple species samples are also necessary to elucidate the interspecific differentiation of *Pedicularis* species.

### 3.3. Population Dynamics Changes

The results of our study indicated that the main suitable habitat of *Pedicularis* is currently located in the Qinling Mountains and adjacent areas. Meanwhile, *P. bicolor* and *P. dissecta* have occurred mainly in eastern Sichuan, Guizhou, and Guangxi during the LIG. In addition, the geographic areas of *P. giraldiana* were also largely distributed in the Yunnan–Guizhou plateau, as previously studied by the transition patterns of the genus *Notopterygium* species [45]. The ENM revealed that three *Pedicularis* species reduced their geographic ranges to the Qinling Mountains and surrounding areas from the LIG to LGM, which coincided with the cooling climate [46,47]. Some other studies also showed that the geological and climatic changes have profoundly affected the geographic distributions and species shifts in the Qinling Mountains and adjacent areas. For example, Jiang et al. (2023) used species distribution modeling to investigate the geographic range changes of an endangered herb *Primula filchnerae* R. Knuth in the Qinling Mountains, and they found that the species range would migrate to the north of China in the future climatic scenario [48]. Meanwhile, Zhao et al. (2023) detected the geographic changes of a shrub species *Bashania fargesii* (E. G. Camus) P. C. Keng and T. P. Yi in the Qinling Mountains, and the authors have also concluded that the species would migrate to the north of China in the future [49]. Additionally, Zhao et al. (2016) detected the geographic shifts of an endangered alpine tree species *Larix chinensis* Beissn. on Taibai Mountain in the Qinling Mountains based on Maxent models, and the authors revealed that the suitable species habitat would move to higher elevations in the future [50].

The main climatic contributions of *Pedicularis* also demonstrated that temperature has a greater effect on the geographic distributions than the precipitation variable (Figure 6). The range shift remains comparatively stable after the glacial periods when the climate was sufficiently warm. When combined with the results of centroid changes, it was clear that three *Pedicularis* species migrated northward during the Quaternary (Figure 5). Generally, climate warming causes the distribution area of many plant species to shift northward in the Northern Hemisphere [51]. This pattern has also been observed in other plants, such as *Roscoea humeana* Balf. f. and W. W. Sm. [52], *Panax notoginseng* (Burkill) F. H. Chen ex C. H. Chow [53], and *Ziziphus jujuba* Mill. [54].

### 3.4. Climatic Niche Comparisons

Under the combined effects of orogenesis and climate changes, species formation and ecological niche differentiation are promoted [55]. The interplay of topographic uplift and climate change in the Himalaya–Hengduan Mountains played a key role in speciation and differentiation during the Quaternary [6]. For example, ecological speciation was demonstrated in two parapatric sister species, *Roscoea cautleoides* Gagnep. and *R. humeana* [56]. Additionally, the heat energy and water in *Pinus yunnanensis* Franch. in the mountain areas promoted intraspecific genetic divergence [57].

Our niche analysis of the environment variables suggested that the three species of *Pedicularis* might have occupied analogous climate niches, with some niche overlaps according to simulations with a 20% occurrence rate (Figure 6). Furthermore, *P. bicolor* and *P. dissecta* inhabited the drier and colder climatic niches, while *P. giraldiana* occupies a hotter and more humid ecological niche, which echoes the previous analysis of the contribution of ecological factors (Figure 3). Additionally, ecological factors, such as temperature and precipitation seasonality, have a significant impact on phenology, including flowering time and growing season [58]. Phenology may have caused gene flow and/or genetic exchange between populations, resulting in population differentiation [59]. Therefore, further research needs to combine population genetic data and environmental selection variables, which have important implications for genetic differentiation and origin between lineages or species, so as to better protect biodiversity in the context of global climate change.

## 4. Materials and Methods

### 4.1. Plant Materials, DNA Extraction, and Genome Assembly

The leaves of *P. bicolor* and *P. giraldiana* were collected from the Qinling Mountains (108°47′10.13″ N, 33°51′27.78″ E and 107°46′23.19″ N, 33°58′9.58″ E, respectively). The vouchers of plant materials were deposited in the herbarium of Northwest University (Xi’an, China). Total genomic DNA was extracted from silica-dried leaves using a modified cetyltrimethylammonium bromide (CTAB) protocol [60].

Paired-end libraries with a 500 bp insert size were constructed and PE150 sequencing was performed on the Illumina HiSeq 2500 platform. GetOrganelle v1.7.0 was used to assemble the chloroplast genomes [61]. The final assembly graph was visualized using Bandage v0.8.1 to authenticate the automatically generated chloroplast genome [62]. The chloroplast genomes were annotated by PGA v1.0 [63] and then manually revised using Geneious v8.1 [64]. The chloroplast genomes of *P. bicolor* and *P. giraldiana* were deposited in GenBank (accession numbers PP439988 and PP439989, respectively). In addition, the chloroplast genome sequence of *P. dissecta* with accession number NC_056312 in the National Center for Biotechnology Information (NCBI) database was downloaded for subsequent analyses.

### 4.2. Phylogenetic Analysis and Divergence Time Estimation

We constructed a phylogenetic tree to determine the evolutionary relationships of three *Pedicularis* species using eighteen complete chloroplast genome sequences from the NCBI database (Appendix A). The general time reversible (GTR) + gamma model was selected based on jmodeltest v2.1.10 tests [65]. The maximum likelihood (ML) analysis was performed using RAxML v8.1.17 with Lamiaceae as the outgroup and 1000 bootstraps [66]. Maximum parsimony (MP) analysis was conducted in PAUP* v4.0 software according to a heuristic search and the tree bisection–reconnection (TBR) branch swapping option [67]. The robustness of inferences was assessed by bootstrap resampling 1000 random replicates.

In addition, BEAST v2.4.5 was used to estimate the node ages and topological structures for *Pedicularis* species [68]. Referring to a previous study, the crown age of Lamiales and Orobanchaceae was dated at 74 Ma and 47.7 Ma, respectively [38,39]. The earliest fossil records of Lamiaceae date back to the early–middle Oligocene around about 28.4 Ma [40]. We applied a GTR model for nucleotide substitution and the “yule process” prior model with three calibration points. The divergence time was estimated by Markov chain Monte Carlo analysis for 50,000,000 generations. TRACER v1.5 was used to analyze the output files to determine whether the effective sample sizes of all parameters were larger than 200. Chronograms with nodal heights and 95% highest posterior density intervals were generated with TreeAnotator v1.7.5 (the first 5000 trees were discarded as a burn-in) and displayed using FigTree v1.0.

### 4.3. Distribution Data Collection

The geographic distribution data of three species of *Pedicularis* were collected through our field surveys in the Qinling Mountains. In addition, we combined the Chinese Virtual Herbarium (https://www.cvh.ac.cn (accessed on 4 January 2024)), Chinese National Knowledge Infrastructure (CNKI, http://www.cnki.net/ (accessed on 4 January 2024)), National Specimen Information Infrastructure (NSII, http://www.nsii.org.cn accessed on 4 January 2024)), and Global Biodiversity Information Facility (GBIF, https://www.gbif.org/ (accessed on 4 January 2024)) to determine geographic coordinates. After removing redundant points, we obtained 55 distribution points for *Pedicularis*, including 11 points for *P. bicolor*, 21 points for *P. giraldiana*, and 23 points for *P. dissecta* (Appendix A).

### 4.4. Ecological Niche Modeling

Previous studies have suggested that in cases of multicollinearity among environmental variables, certain environmental gradients may have a more pronounced effect on the model outcomes, potentially resulting in greater uncertainty [69,70]. Therefore, environmental variables should be screened to reduce the influence of multicollinearity. The 19 climatic variables were downloaded from the WorldClim database (Appendix A). We examined the correlation between all layers using ENMTools v1.3 [71]. Two variables were considered highly correlated when the correlation coefficient was above 0.8.

We used MaxEnt v3.3.3 to predict the current, last glacial maximum (LGM, 0.021–0.018 Ma), Last Interglacial (LIG, 0.140–0.120 Ma), and future (years 2050 and 2070) potential geographic distributions of three *Pedicularis* species [72]. For future climate variables, we used climate data in the 2050s and 2070s. Two representative concentration pathway scenarios (RCP 2.6 and RCP 8.5) developed by the Intergovernmental Panel on Climate Change (IPCC) were used in this study, with RCP 2.6 used as the minimum emission scenario and RCP 8.5 as the maximum emission scenario [73]. We used the default parameters for MaxEnt and employed the “subsample” method (setting the number of replicates to 20), with 75% of the species records for training and 25% for testing the model. The overall performance of the model was assessed by calculating the area under the curve (AUC). AUC scores range from 0.5 (indicating randomness) to 1 (indicating an exact match), with a value above 0.9 considered indicative of good model performance [74].

### 4.5. Niche Comparison Analyses

Niche comparison analysis was performed to test whether the selection of different types of climatic niches may have contributed to the divergence of three *Pedicularis* species. We assessed ENMs in a spatial environment using R packages ecospat v4.0 [75]. Furthermore, to measure niche differences between species, we used ENMTools v1.3 to calculate the niche overlap statistic Schoener’s *D* and standardized Hellinger’s *I* distance [76,77], where a value of 0 denotes no overlap and 1 indicates complete overlap. To test the null hypothesis that two species have identical ENMs, we used the niche identity test initially in ENMTools v1.3. This test compares the observed scores of niche overlap statistics *D* and *I* with their null distribution generated with 100 pseudo-replicates. Finally, niche overlaps between species were characterized by niche unfilling, niche stability, and niche expansion [78,79].

## Figures and Tables

**Figure 1 plants-13-00765-f001:**
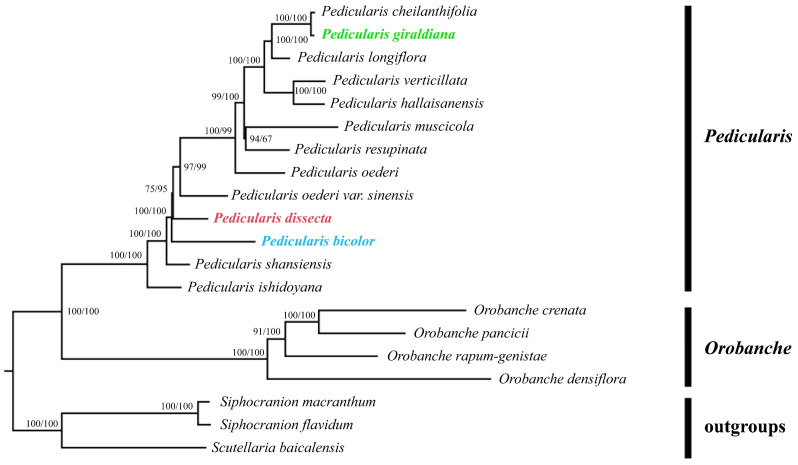
Phylogenetic tree based on chloroplast genomes of *Pedicularis* species. Numbers above the branches are bootstrap values according to maximum likelihood (**left**) and maximum parsimony (**right**) analyses. The three colorful species are main objects of this study.

**Figure 2 plants-13-00765-f002:**
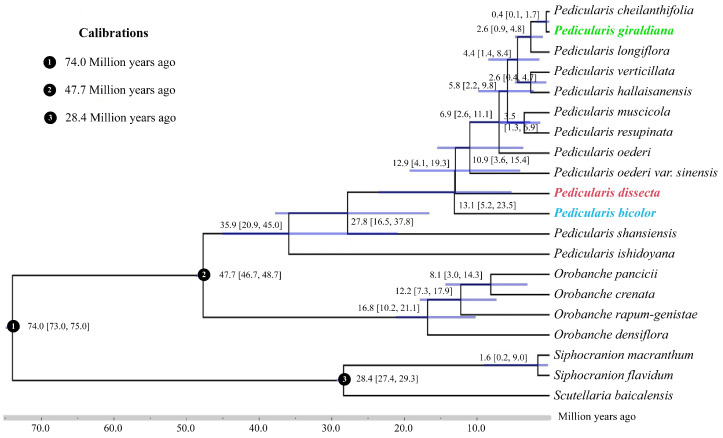
Divergence times based on chloroplast genomes. Blue bars and the numbers below the bars indicate 95% highest posterior densities of divergence times (million years ago). The three colorful species are main objects of this study.

**Figure 3 plants-13-00765-f003:**
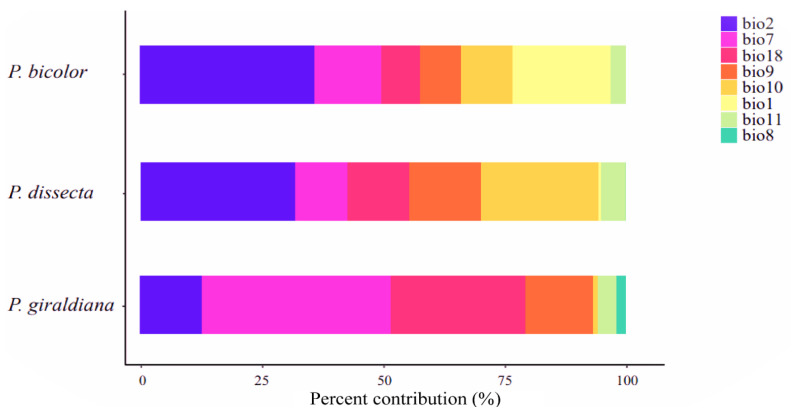
Environmental variable contributions of the ecological niche model for three *Pedicularis* species. Bio1, annual mean temperature; bio2, mean diurnal range; bio7, temperature annual range; bio8, mean temperature of the wettest quarter; bio9, mean temperature of the driest quarter; bio10, mean temperature of the warmest quarter; bio11, mean temperature of the coldest quarter; bio18, precipitation of the warmest quarter.

**Figure 4 plants-13-00765-f004:**
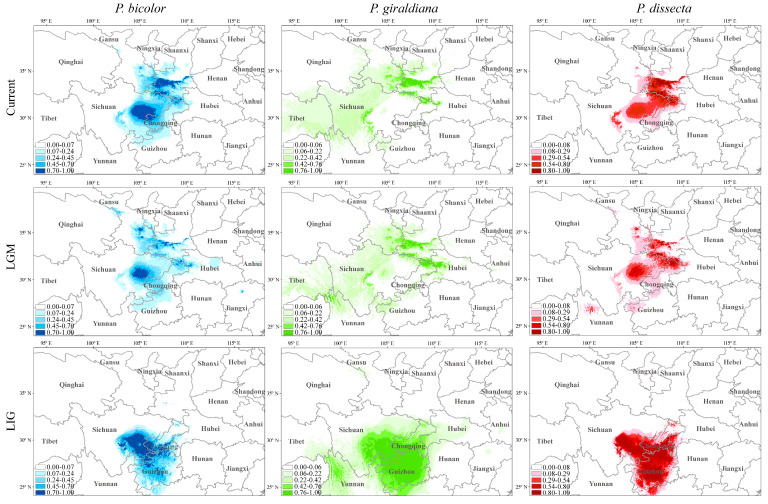
Distribution of three *Pedicularis* species at present, last glacial maximum (LGM), and last interglacial (LIG) based on ecological niche modeling. The light to dark colors and small to large numbers of the small squares in the legend correspond to the low to high suitability of the species for the potential geographic ranges.

**Figure 5 plants-13-00765-f005:**
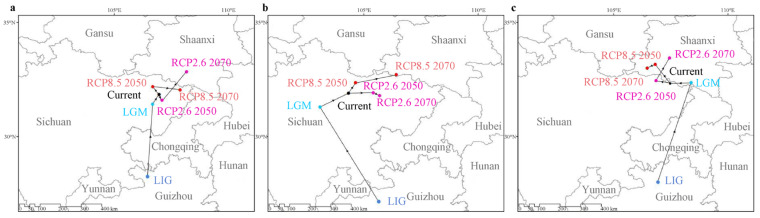
Migration of the center of suitable habitat for three *Pedicularis* species. (**a**) *P. bicolor*; (**b**) *P. giraldiana*; (**c**) *P. dissecta*. LGM, last glacial maximum; LIG, last interglacial; RCP, representative concentration pathway scenario. RCP2.6 and RCP8.5 are defined according to the radiative forcing target level from 2.6 W/m^2^ (low carbon emissions) to 8.5 W/m^2^ (high carbon emissions) for 2100.

**Figure 6 plants-13-00765-f006:**
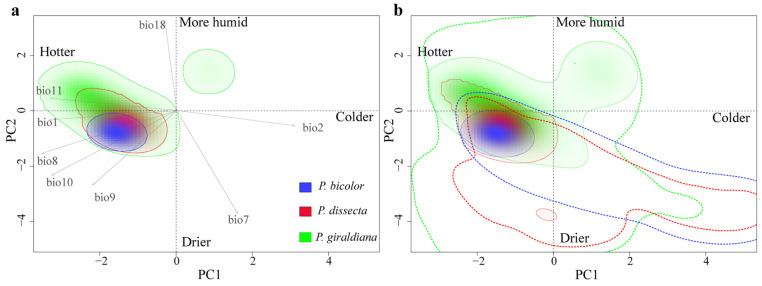
Geographic location of the occurrence records and background areas used for the climatic niche comparison in the e-space of three *Pedicularis* species: *P. bicolor* (blue), *P. dissecta* (red), and *P. giraldiana* (green). (**a**,**b**) represent climatic space through the environmental principal components analysis (PCA-env) (explaining PC1 = 70.3% and PC2 = 18.6% of the total climatic variation). In (**a**), the solid lines delimit the 20% of occurrence density of the current occupied niches by the species. Gray arrows outline the direction of the variables’ contribution to the PCA-env. In (**b**), the solid and dashed lines illustrate the 100% occurrence density and the 100% available background climates, respectively. Bio1, annual mean temperature; bio2, mean diurnal range; bio7, temperature annual range; bio8, mean temperature of the wettest quarter; bio9, mean temperature of the driest quarter; bio10, mean temperature of the warmest quarter; bio11, mean temperature of the coldest quarter; bio18, precipitation of the warmest quarter.

**Figure 7 plants-13-00765-f007:**
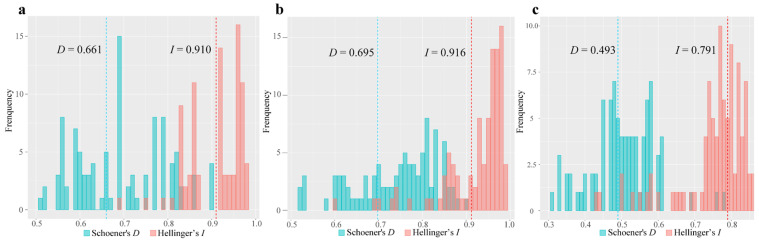
Niche identity test for each comparison based on Schoener’s *D* statistic and Hellinger’s *I* statistic. (**a**–**c**) represent the comparison results between *P. bicolor* vs. *P. dissecta*, *P. dissecta* vs. *P. giraldiana*, and *P. giraldiana* vs. *P. bicolor*, respectively. Bars indicate the null distributions of *D* and *I*.

**Table 1 plants-13-00765-t001:** Comparisons of the current occupied climatic niches between *Pedicularis* species.

	Niche Overlap	Niche Unfilling	Niche Stability	Niche Expansion
*P. bicolor—P. dissecta*	0.463	0.036	0.624	0.376
*P. dissecta—P. giraldiana*	0.204	0.126	0.427	0.573
*P. giraldiana—P. bicolor*	0.122	0.000	0.276	0.724

## Data Availability

The chloroplast genomes of *P. bicolor* and *P. giraldiana* were deposited in GenBank (accession numbers PP439988 and PP439989, respectively).

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
