# Peer review of "Responses of Three Pedicularis Species to Geological and Climatic Changes in the Qinling Mountains and Adjacent Areas in East Asia"

_plants, 2024, doi:10.3390/plants13060765_

Round 1

Reviewer 1 Report

Comments and Suggestions for Authors

Author Response

Dear Reviewer,

Thank you very much for kindly comments and suggestions. We are very grateful to you for your time and patience, as well as your constructive and thoughtful comments. These suggestions and comments are all valuable and very helpful for revising and improving our MS. In the revised manuscript, we have studied all these comments carefully and have made thoroughly corrections, and responded point by point to the comments as itemized below (our responses are in bold characters).

Comments and Suggestions for Authors

Line 92: Add the patronymic of the three Pedicuaris species and add the reference  POWO

Reply: Thank you very much for your very helpful and constructive comments and suggestions. In the revised manuscript, we have revised the genus and nomenclator of the three Pedicuaris species accordingly.

Line 102: The first paragraph  " In this study.....Last Glacial Maximum (LGM)  is redundant respect the second paragraph. Delete this first paragraph.

Reply: Thank you very much again for your kindly suggestions and comments. In the revised manuscript, we have deleted the last glacial maximum (LGM) in the first paragraph accordingly.

The quality of image needs to be greatly improved. It is clearly not readable.

Reply: Many thanks for your kindly comments. In the revised manuscript, we have improved the clarity of all the images in the manuscript and enlarged the text in the figures.

What is the temperature variation? Please add a range of values for the precipitations, low temperatures.

Reply: Thank you for your helpful comments. We have revised the error in the description of this sentence accordingly. By comparing the contribution of different climatic variables, we found that mean diurnal range and temperature annual range plays key roles in the geographical distributions of three Pedicularis species. Therefore, we would like to suggest that the changes of diurnal and annual temperature are important of geographic distributions for three Pedicularis species.

Line 157-158: All the climatical variables from bio1 to bio11 and bio18 must be reported in the caption with the explanation.

Reply: Many thanks for your kindly comments. We have added the detailed descriptions of each climate variable below the figures in the text accordingly.

What kind of emission in line 169?

Reply: Thank you for your helpful comments. We have revised the sentence on emissions in the manuscript to illustrate potential regional changes in the geographic distributions of the three Pedicularis species under two future representative concentration pathway scenarios (RCP2.6 and RCP8.5). RCP2.6 and RCP8.5 were defined according to the radiative forcing target level from 2.6 W/m2 to 8.5 W/m2 for 2100.

What is the origin of P. dissecta materials? The information is not reported.

Reply: Many thanks for your kindly comments. In the revised manuscript, we have added the information of P. dissecta in the revised manuscript. The chloroplast genome of P. dissecta (accession number: NC_056312) was downloaded from the National Center for Biotechnology Information (NCBI) and used for subsequent analyses.

Line 368: Add the reference for the NCBI database

Reply: Many thanks for your helpful comments. We have added the references in table S1.

Line 400: This selection of the variables using ENMTools is a result, so you have to move from Materials and methods.

Reply: Thank you for your helpful comments. We have moved these parts from materials and methods accordingly.

Reviewer 2 Report

Comments and Suggestions for Authors

Plants, Febr. 2024

Article: Responses of three Pedicularis

Authors: Zhang Q. et al.

Comments for the authors

General remark

It has been made possible by the authors to clarify the evolutionary origin and the phylogenetic relationship of three Pedicularis species growing in the Quinling Mountains and adjacent areas in China. Secondly, also the bold attempt to describe the geographic distributions of these three species at the last glacial maximum period and the current period, and even to predict their performance under future climate change has led to success by modeling. Therefore, in the opinion of the reviewer, this study fulfills the criteria of originality and novelty.

It must be emphasized in addition, that the three selected species are important for nature protection.

Unfortunately, the authors did not fully enable the readers to understand their contribution to scientific progress because they did not present their figures in a quality which is mostly considered to be necessary. The same applies to numerous abbreviations. (Compare ‘Detailed Comments’!

This reviewer is not an expert for English. However, sometimes it seems as if the authors did not involve a native English-speaking person but used a language program (especially for the ‘Abstract’).

These critics should be considered during revision of this first version.

Detailed comments

Page 1

Abstract

Lines 20-21: ‘… in the Quinling area in East Asia …’ Replace it by ‘… there …’!

Line 22: Perhaps better English: Instead of ‘… of three Pedicularis …’ ‘…for three Pedicularis …’.

Line 26: English? ‘… was take place …’?  This reviewer is not an expert in English. Nevertheless, he feels that this is not correct.

Line 30: ‘… (bio2) … and ‘… (bio7) … are not necessary in the ‘Abstract’ and should be omitted.

Line 32: ‘… have the similarity ecological range …’ English? (similar?)

Introduction

Line 43: ‘…of plants …’

Page 2

Line 46: ‘… many extant …’? Meaning?

Line 90: Probably: ‘… vulnerable …’ (Written with a lower-case letter!)

Page 3

Results

Figure 1: It is recommended to enlarge the numbers and the plant names in the graphs for improving readability.

Why do the authors use lower case letters for ‘… maximum likelihood …’ and ‘… maximum parsimony …’. In contrast, in the text of the manuscript ‘M…’ and ‘L…’ and ‘M…’ and ‘P…’ is used. Please, stay uniform!

Lines 124-129: Are two digits behind the point realistic (47.72 etc.) or would it not be better to write ‘… ~47.7 etc.? Can this be measured so exactly?

Page 4

Figure 2: Again, all numbers and names (inclusive those in the box) are too tiny and might be lost in print.

Figure 3: ‘bio2, bio7, etc. should be briefly explained in the text (legend) underneath the figure. (A general rule says that figures and tables should be understandable by themselves independently from reading the complete text of the manuscript.)

Page 5

Figure 4: Also, this figure cannot be understood independently. The abbreviations along the y-axis are not explained in the text (legend) underneath the figure. The legends in the sections of the graph, explaining the colors, are not at all readable. In addition, some of the Chinese province names should be visibly written into the maps.

Page 6

Figure 5:  Again, the legend underneath the graph does not explain the inscriptions and colors within the graphs. Secondly, names and numbers in the graphs are too delicate and tiny. (The size of the graphs could be maintained but letters and numbers should better be enlarged if one wants to supply the reader with some information.)

Figure 6: Again, this figure needs to be enlarged as far as numbers and inscriptions are concerned. Secondly, abbreviations must be avoided in the text (legend) underneath the graph. Thirdly, ‘bio1’, ‘bio8’ …etc. need explanation in the text underneath the graph. (It would be a pity, if this beautiful figure would be wasted because it cannot be understood by readers.)

Page 7

Figure 7: Compare all the comments for the other graphics. Here, especially the inscriptions along the axes are not readable.

Discussion

Lines 234-236: It is not necessary to repeat ‘Material and Methods’ in the chapter ‘Discussion’. Therefore, this sentence can be omitted.

Page 8

Lines 255-256: It is not necessary to repeat ‘Material and Methods’ in the chapter ‘Discussion’.

Lines 260, 261, 264, 270, 275: In the relevant literature one digit behind the point seems to be enough Therefore, it is recommended that authors write ~35.9 (or 35.9 only) etc.

Lines 283-288: This belongs to ‘Introduction’ and should be deleted. One could start with line 289: ‘The results of our study …’.

Page 9

Material and Methods

Line 331: What means the abbreviation ‘CTAB’?

Line 340: What means the abbreviation ‘NCBI’? What means ‘GTR + G model’?

Line 343: What means ‘PAUP*v4.0’?

Line 344: What means ‘TBR branch’.  Please supply us with full names!

Line 386: ‘AUC’?

References

There are many cross references. Citations were only checked at random by this reviewer. It is recommended that the authors control them all again.

Final statement

In the opinion of this reviewer, it is worthwhile to improve this manuscript thoroughly and submit it again for publication.

Comments on the Quality of English Language

This reviewer is not an expert for English. However, he found some inaccurancies (Compare 'Comments for the authors'!)

Author Response

Dear Reviewer,

Thank you very much for kindly comments and suggestions. We are very grateful to you for your time and patience, as well as your constructive and thoughtful comments. These suggestions and comments are all valuable and very helpful for revising and improving our MS. In the revised manuscript, we have studied all these comments carefully and have made thoroughly corrections, and responded point by point to the comments as itemized below (our responses are in bold characters).

Comments and Suggestions for Authors

General remark

It has been made possible by the authors to clarify the evolutionary origin and the phylogenetic relationship of three Pedicularis species growing in the Qinling Mountains and adjacent areas in China. Secondly, also the bold attempt to describe the geographic distributions of these three species at the last glacial maximum period and the current period, and even to predict their performance under future climate change has led to success by modeling. Therefore, in the opinion of the reviewer, this study fulfills the criteria of originality and novelty.

Reply: Thank you very much for your kindly comments and suggestions for our manuscript. We are also grateful to you for your positive evaluations. Additionally, in the revised manuscript, we have studied the all comments carefully and have made thoroughly corrections according to your kindly and helpful comments and suggestions, we hope that this version is suitable to the journal. Thank you very much again for your kindly comments.

It must be emphasized in addition, that the three selected species are important for nature protection.

Reply: We thank you for your positive comments for our manuscript. We also highly agree with your ideas that the three selected species are important for nature protection.

Unfortunately, the authors did not fully enable the readers to understand their contribution to scientific progress because they did not present their figures in a quality which is mostly considered to be necessary. The same applies to numerous abbreviations. (Compare ‘Detailed Comments’!

Reply: Many thanks for your constructive and helpful comments and suggestions for our manuscript. In the revised MS, we have re-figured the all figures and made them clearer. Meanwhile, we have also made thoroughly all abbreviations and other parts accordingly.

This reviewer is not an expert for English. However, sometimes it seems as if the authors did not involve a native English-speaking person but used a language program (especially for the ‘Abstract’).

Reply: Thanks for your kindly comments and suggestions. In the revised version, we have modified all the parts carefully and thoroughly checked and revised all the English grammar accordingly. We hope that this version is suitable to the journal.

Detailed comments

Page 1

Abstract

Lines 20-21: ‘… in the Qinling area in East Asia …’ Replace it by ‘… there …’!

Reply: Thank you very much for your very helpful and constructive comments and suggestions. In the revised manuscript, we have replaced ‘in the Qinling area in East Asia’ by ‘there’ in the sentence accordingly.

Line 22: Perhaps better English: Instead of ‘… of three Pedicularis …’ ‘…for three Pedicularis …’.

Reply: Thank you very much for your helpful suggestions. In the revised manuscript, we have revised these parts accordingly.

Line 26: English? ‘… was take place …’?  This reviewer is not an expert in English. Nevertheless, he feels that this is not correct.

Reply: Many thanks for your kindly comments. In the revised manuscript, we have carefully and thoroughly corrected the errors in the manuscript accordingly.

Line 30: ‘… (bio2) … and ‘… (bio7) … are not necessary in the ‘Abstract’ and should be omitted.

Reply: Thank you very much for your helpful suggestions. We have deleted bio2 and bio7 in the abstract accordingly.

Line 32: ‘… have the similarity ecological range …’ English? (similar?)

Reply: Thank you for your helpful comments. In the revised manuscript, we have revised this sentence accordingly.

Introduction

Line 43: ‘…of plants …’

Reply: Many thanks for your kindly comments. In the revised manuscript, we have carefully corrected this mistake in the manuscript.

Line 46: ‘… many extant …’? Meaning?

Reply: Thank you for your helpful comments. Quaternary climatic fluctuations have influenced species differentiation and distributional ranges (e.g. Kalopanax septemlobus and endangered Saruma henryi in references 3 and 4). In the revised manuscript, we have modified this sentence accordingly, please see in the text.

Line 90: Probably: ‘… vulnerable …’ (Written with a lower-case letter!)

Reply: Many thanks for your kindly advice. We checked and corrected both upper and lower case in the manuscript accordingly.

Page 3

Results

Figure 1: It is recommended to enlarge the numbers and the plant names in the graphs for improving readability.

Reply: Many thanks for your kindly comments. In the revised manuscript, we have improved the clarity of all the figures in the manuscript and enlarged the text in the graphs.

Why do the authors use lower case letters for ‘… maximum likelihood …’ and ‘… maximum parsimony …’. In contrast, in the text of the manuscript ‘M…’ and ‘L…’ and ‘M…’ and ‘P…’ is used. Please, stay uniform!

Reply: Thank you for your helpful comments. We have checked and revised these parts in the manuscript accordingly.

Lines 124-129: Are two digits behind the point realistic (47.72 etc.) or would it not be better to write ‘… ~47.7 etc.? Can this be measured so exactly?

Reply: Thank you for your constructive comments. In the revised manuscript, we carefully modified all numbers of divergence times in figure 2 accordingly.

Figure 2: Again, all numbers and names (inclusive those in the box) are too tiny and might be lost in print.

Reply: Thank you very much for your advice. In the revised manuscript, we have carefully and thoroughly changed the clarity of all the figures in the manuscript accordingly.

Figure 3: ‘bio2, bio7, etc. should be briefly explained in the text (legend) underneath the figure. (A general rule says that figures and tables should be understandable by themselves independently from reading the complete text of the manuscript.)

Reply: Thank you very much for your helpful suggestions. We have added the brief explanations of each climate variable in the text accordingly.

Figure 4: Also, this figure cannot be understood independently. The abbreviations along the y-axis are not explained in the text (legend) underneath the figure. The legends in the sections of the graph, explaining the colors, are not at all readable. In addition, some of the Chinese province names should be visibly written into the maps.

Reply: Many thanks for your kindly comments. In the revised manuscript, we have improved the clarity of all the images in the manuscript and enlarged the text in the figures. We have also added the explanations of the abbreviations below all the figures in the text. In addition, we have also added the names of some Chinese provinces to the figures accordingly.

Figure 5: Again, the legend underneath the graph does not explain the inscriptions and colors within the graphs. Secondly, names and numbers in the graphs are too delicate and tiny. (The size of the graphs could be maintained but letters and numbers should better be enlarged if one wants to supply the reader with some information.)

Reply: Thank you for your helpful comments. We have explained the abbreviations in the legend underneath figure 5 and enlarged the letters and numbers accordingly.

Figure 6: Again, this figure needs to be enlarged as far as numbers and inscriptions are concerned. Secondly, abbreviations must be avoided in the text (legend) underneath the graph. Thirdly, ‘bio1’, ‘bio8’ …etc. need explanation in the text underneath the graph. (It would be a pity, if this beautiful figure would be wasted because it cannot be understood by readers.)

Reply: Thank you very much for your advice. In the revised manuscript, we have enlarged the size of the letters and numbers, and also changed the abbreviation of the legend. In addition, we have also added the descriptions of biological variables in the text.

Figure 7: Compare all the comments for the other graphics. Here, especially the inscriptions along the axes are not readable.

Reply: Thank you very much for your helpful suggestions. In the revised manuscript, we have enlarged all the annotations in figure 7.

Discussion

Lines 234-236: It is not necessary to repeat ‘Material and Methods’ in the chapter ‘Discussion’. Therefore, this sentence can be omitted.

Reply: Thank you for your helpful comments. We have removed these parts accordingly.

Lines 255-256: It is not necessary to repeat ‘Material and Methods’ in the chapter ‘Discussion’.

Reply: Thank you very much for your helpful and constructive comments and suggestions. We have deleted the method of estimating differentiation times using the BEAST software in the discussion. We added the calibrations and references in the section 4.2 accordingly.

Lines 260, 261, 264, 270, 275: In the relevant literature one digit behind the point seems to be enough Therefore, it is recommended that authors write ~35.9 (or 35.9 only) etc.

Reply: Thank you very much for your helpful suggestions. In the revised manuscript, we carefully modified all numbers of divergence times in figure 2 accordingly.

Lines 283-288: This belongs to ‘Introduction’ and should be deleted. One could start with line 289: ‘The results of our study …’.

Reply: Thank you very much for your helpful suggestions. We have revised these parts accordingly.

Page 9

Material and Methods

Line 331: What means the abbreviation ‘CTAB’?

Reply: Thank you for your helpful comments. We have added the full name of CTAB to the revised manuscript accordingly.

Line 340: What means the abbreviation ‘NCBI’? What means ‘GTR + G model’?

Reply: Thank you very much for your helpful comments. In the revised manuscript, we have added the descriptions of NCBI and GTR + G model. NCBI is National Center for Biotechnology Information. GTR + G model is general time reversible + gamma, a model for phylogenetic analysis. In addition, we have added the full names of abbreviations for other parts.

Line 343: What means ‘PAUP*v4.0’?

Reply: Thank you very much for your kindly comments. In the revised manuscript, we have added the description of PAUP*v4.0. PAUP*v4.0 is a software for constructing phylogenetic trees. Additionally, we have added the descriptions of abbreviations for other parts.

Line 344: What means ‘TBR branch’.  Please supply us with full names!

Reply: Thank you very much for your kindly comments. In the revised manuscript, we have added the full name of TBR accordingly. Meanwhile, we have added explanations of abbreviations for other methods.

Line 386: ‘AUC’?

Reply: Thank you for your helpful comments. In the revised manuscript, we have added the full name of AUC.

Reviewer 3 Report

Comments and Suggestions for Authors

Dear authors,

You are presenting an interesting paper concerning “Responses of three Pedicularis species to geological and climatic changes in the Qinling Mountains and adjacent areas in East Asia”.  I like the overall idea and the methodology of your paper that gives interesting results. The manuscript is well written and structured. Discussion could be enriched with more comparisons of your results with those of other studies investigating responses of species to environmental abiotic changes. This version of the manuscript could be much improved.  I believe that a revision of the MS is needed so that the science presented, which is interesting, can be communicated more effectively.

Please find below a few minor comments.

Page 3, lines 123-129: Please give explanations for BEAST, HPD.

Page 4, line 135: Please give explanations for all of “bio1, bio2, bio7, bio8, bio9, bio10, bio11 and bio18”. Only bio2, bio7, bio9 and bio 10 are described below.

Page 7: Table 1 is mentioned in the text after its appearance. Please correct.

Figure 7: Letters in the diagrams are very small and cannot easily be read. Please make them larger.

I hope that my comments will be helpful.

Author Response

Dear Reviewer,

Thank you very much for kindly comments and suggestions. We are very grateful to you for your time and patience, as well as your constructive and thoughtful comments. These suggestions and comments are all valuable and very helpful for revising and improving our MS. In the revised manuscript, we have studied all these comments carefully and have made thoroughly corrections, and responded point by point to the comments as itemized below (our responses are in bold characters).

Comments and Suggestions for Authors

You are presenting an interesting paper concerning “Responses of three Pedicularis species to geological and climatic changes in the Qinling Mountains and adjacent areas in East Asia”. I like the overall idea and the methodology of your paper that gives interesting results. The manuscript is well written and structured. Discussion could be enriched with more comparisons of your results with those of other studies investigating responses of species to environmental abiotic changes. This version of the manuscript could be much improved. I believe that a revision of the MS is needed so that the science presented, which is interesting, can be communicated more effectively.

Reply: Thank you very much for your kindly comments and suggestions for our manuscript. We are also grateful to you for your positive evaluations. In the revised manuscript, we have studied the all comments carefully and have made thoroughly corrections according to your kindly and helpful comments and suggestions. We have also enriched with more comparisons of our results with those of other studies investigated the responses of species to environmental changes. We hope that this version is suitable to the journal. Thank you very much again for your kindly comments.

Please find below a few minor comments.

Page 3, lines 123-129: Please give explanations for BEAST, HPD.

Reply: Thank you very much for your very helpful and constructive comments and suggestions. In the revised manuscript, we have revised these parts accordingly.

Page 4, line 135: Please give explanations for all of “bio1, bio2, bio7, bio8, bio9, bio10, bio11 and bio18”. Only bio2, bio7, bio9 and bio 10 are described below.

Reply: Many thanks for your suggestions. In the revised manuscript, we have carefully and revised the explanations for all of bioclimatic variables (bio1, bio2, bio7, bio8, bio9, bio10, bio11 and bio18).

Page 7: Table 1 is mentioned in the text after its appearance. Please correct.

Reply: Many thanks for your kindly comments. We have adjusted the position of table 1 in the revised manuscript.

Figure 7: Letters in the diagrams are very small and cannot easily be read. Please make them larger. 

Reply: Thank you very much for your helpful comments and suggestions. In the revised manuscript, we have improved the clarity of all the figures in the manuscript accordingly.

Round 2

Reviewer 3 Report

Comments and Suggestions for Authors

Dear authors,

The new version of the manuscript is much updated.